# Prevalence and molecular heterogeneity of glucose-6-phosphate dehydrogenase (G6PD) deficiency in the Senoi Malaysian Orang Asli population

Danny Xuan-Rong Koh[1], Mohamed Afiq Hidayat Zailani[2], Raja Zahratul Azma Raja Sabudin[2]*, Sanggari Muniandy[1], Nur Awatif Akmal Muhamad Hata[3], Siti Noor Baya Mohd Noor[3], Norhazilah Zakaria[3], Ainoon Othman[4], Endom Ismail[5]

1 Faculty of Science and Technology, Center of Frontier Sciences, Universiti Kebangsaan Malaysia, Selangor, Malaysia, 2 Faculty of Medicine, Department of Pathology, Universiti Kebangsaan Malaysia (UKM), Kuala Lumpur, Malaysia, 3 Faculty of Medicine, Department of Diagnostic Laboratory Services, Universiti Kebangsaan Malaysia (UKM), Kuala Lumpur, Malaysia, 4 Faculty of Medicine and Health Sciences, Department of Pathology, Universiti Sains Islam Malaysia, Negeri Sembilan, Malaysia, 5 Faculty of Science and Technology, Department of Biological Sciences Dan Biotechnology, Universiti Kebangsaan Malaysia, Selangor, Malaysia

* zahratul@ppukm.ukm.edu.my

## Abstract

Glucose-6-phosphate dehydrogenase (G6PD) deficiency is an X-linked genetic disorder characterized by reduced G6PD enzyme levels in the blood. This condition is common in populations exposed to malaria; an acute febrile disease caused by *Plasmodium* parasites. G6PD-deficient individuals may suffer from acute hemolysis following the prescription of Primaquine, an antimalarial treatment. The population at risk for such a condition includes the Senoi group of Orang Asli, a remote indigenous community in Malaysia. This study aimed to elucidate the G6PD molecular heterogeneity in this subethnic group which is important for malaria elimination. A total of 662 blood samples (369 males and 293 females) from the Senoi subethnic group were screened for G6PD deficiency using a quantitative G6PD assay, OSMMR2000-D kit with Hb normalization. After excluding the family members, the overall prevalence of G6PD deficiency in the studied population was 15.2% (95% CI: 11–19%; 56 of 369), with males (30 of 172; 17.4%) outnumbering females (26 of 197; 13.2%). The adjusted male median (AMM), defined as 100% G6PD activity, was 11.8 IU/gHb. A total of 36 participants (9.6%; 26 male and 10 female) were deficient (<30% of AMM) and 20 participants (5.4%; 4 male and 16 female) were G6PD-intermediate (30–70% of AMM). A total of 87 samples were genotyped, of which 18 showed no mutation. Seven mutations were found among 69 genotyped samples; IVS11 T93C (47.1%; n = 41), rs1050757 (3'UTR +357A>G)(39.1%; n = 34), G6PD Viangchan (c.871G>A)(25.3%; n = 22), G6PD Union (c.1360C>T)(21.8%; n = 19), c.1311C>T(20.7%; n = 18), G6PD Kaiping (c.1388G>A) (8.0%; n = 7), and G6PD Coimbra (c.592C>T)(2.3%; n = 2). Our analysis revealed 27 hemizygote males, 18 heterozygote females, 7 homozygote females, and 2 compound heterozygote females. This study confirms the high prevalence of G6PD deficiency among the Senoi Malaysian Orang Asli, with a significant degree of molecular heterogeneity. More emphasis

**Data Availability Statement:** All relevant data are within the paper and its Supporting Information files.

**Funding:** RZA received the award that funded this study. This study was funded by the Ministry of Higher Education (MOHE), Malaysia through the Exploratory Research Grant Scheme (Grant number: ERGS/1/2012/STG03/UKM/02/1). The website of funder is https://www.mohe.gov.my/. The funders had no role in study design, data collection and analysis, decision to publish, or preparation of the manuscript.

**Competing interests:** The authors have declared that no competing interests exist.

should be placed on screening for G6PD status and proper and safe use of Primaquine in the elimination of malaria among this indigenous population.

## Introduction

Glucose-6-phosphate dehydrogenase (G6PD) deficiency is a common genetic disorder that affects more than 500 million people worldwide [1]. G6PD acts as a key house-keeping enzyme that oxidizes glucose-6-phosphate to 6-phosphogluconolactone in the pentose phosphate pathway and reduces nicotinamide adenine dinucleotide phosphate (NADP) to nicotinamide adenine dinucleotide phosphate-oxidase (NADPH), a crucial reducing agent in the protection of red blood cells against oxidative stress [1]. This X-linked hereditary condition affects the entire world, but it is more prevalent, particularly in parts of the African continent, the Middle East, and Southeast Asia, where it often overlaps with the geographical distribution of malaria infection [2].

In Malaysia, the overall prevalence of G6PD deficiency among males was 4.7%, which was detected in three main ethnics in Malaysia including Chinese (6.0%) which was the most predominant, followed by Malays (4.6%) and Indian (1.3%) [3,4]. The Orang Asli was the indigenous population of Peninsular Malaysia that formed a national minority. They were formed of three groups, namely Senoi, Proto Malay, and Negrito, which were further divided into six different subethnics in each group [5].

The Senoi population represents the largest group of the Orang Asli in Malaysia (54.9%), followed by the Proto Malay (42.3%) and Negrito (2.8%) [6]. These indigenous groups inhabit remote areas including the tropical rainforest and traditionally adopt a hunter-gatherer lifestyle with self-subsistence culture such as jungle farming. Their lifestyle habit and geographical distributions both resulted in constant exposure to neglected tropical and infectious diseases such as malaria and soil-transmitted helminth infection among the Orang Asli population [7,8]. A previous epidemiological study revealed that the endemicity of malaria among the Orang Asli population was 24.2%, with at least 3.7 times more in the Orang Asli children under 12 years [9].

Malaria is an acute febrile infection caused by *Plasmodium* parasites. It is a medical emergency as the infection may rapidly progress to life-threatening complications such as cerebral malaria, pulmonary edema, acute renal failure, and severe anemia. If the infected individuals do not receive prompt and appropriate treatment, these conditions can be fatal [10,11]. Currently, Primaquine is the most widely used anti-malarial drug in the world [12,13], capable of treating and eliminating *P. vivax* liver stage infection and gametocytocidal activity for *P. falciparum* [14]. However, in G6PD-deficient patients, this medication may cause oxidative stress resulting in acute hemolysis.

The Orang Asli population is highly susceptible to mosquito bites and malaria infection due to their isolated settlements in tropical forests and traditions of hunting and foraging for food in the jungle. Despite the high susceptibility and endemicity of malaria among the Orang Asli in Malaysia, there is a scarcity of molecular studies of the G6PD variants that may impede a successful radical cure of malaria infection.

Iwai et al. (2001) and Wang et al. (2008) were among the first to report cases of G6PD deficiency among the Orang Asli [14,15]. However, both studies did not provide detailed subethnicity information. A study was conducted among Temiar, a subethnic group of the Senoi Orang Asli, and discovered that the population's G6PD incidence was 52.4% (36.2% in males,

29.8% in females) [16]. Another study which was conducted on the Negrito, the smallest Orang Asli group showed that the ethnicity had a 9% prevalence, with the highest incidence coming from the Lanoh subethnic group of Negrito Orang Asli (28%) [17]. However, these studies were conducted using a fluorescent spot test (FST), a qualitative method that was proven to have a lack of sensitivity to detect G6PD deficiency, particularly in females heterozygotes and those with moderate enzyme activity ranging between 20 to 60% of the normal mean [3,18,19].

In addition, a previous study successfully identified three novel single nucleotide polymorphisms (SNP) in the 3' untranslated region (3'UTR) among the deficient Negritos, which were rs112950723, rs111485003, and rs1050757G [20]. Further investigation on these SNPs revealed that only the rs1050757G significantly changed the secondary structure of the mutant transcript, whereas the rs112950723 and rs111485003 did not affect mRNA folding [21]. Nonetheless, according to this study, additional experimental research is needed to reliably determine the role of mRNA secondary structure on G6PD deficiency. This study aimed to characterize the genetic profile of the Senoi Orang Asli and to elucidate the G6PD molecular heterogeneity in this subethnic group.

## Materials and methods

### Study design

This was a cross-sectional study involving all subethnic groups of the Senoi Orang Asli population in Peninsular Malaysia, namely Che Wong, Mah Meri, Jah Hut, Semaq Beri, Semai, and Temiar. The ethical approval for this study was obtained from the Universiti Kebangsaan Malaysia (UKM) Medical Centre Ethics Committee (UKM1.5.3.5/244) and the Malaysian Department of Orang Asli Development (JHEOA.PP.30.052Jld.6(13)).

A total of 662 consenting volunteers consisting of 369 males and 293 females of the Senoi subethnic were enrolled in this study. The inclusion criteria were healthy Senoi Orang-Asli individuals of pure, single subethnic lineage for three generations. Individuals with an unclear or mixed subethnics lineage as well as those with any systemic or blood disease were excluded from this study. All participants were informed about the nature of the study, and written consent was obtained from each individual prior to sample collection.

### Sample collection and processing

Six ml venous blood sample was collected from each participant in ethylene diamine tetra acetic acid (EDTA) tubes using sterile venipuncture techniques. The blood samples were stored in an insulated container with ice packs and were sent to the laboratory at the Haematology Unit, Department of Diagnostic Laboratory Services (JPMD), UKM Medical Centre (UKMMC) within 24 hours post-collection.

In the laboratory, the G6PD activity of the samples was measured using a quantitative method, OSMMR2000-D assay kit with hemoglobin (Hb) normalization (OSMMR; R&D Diagnostics, N. Dimopoulos S.A, Greece). Five microliters of the blood sample were mixed with 75 μl of elution buffer. The mixture and 75 μl of the kit's reagent were put in a separate well of a microplate. This microplate preparation was then steadily warmed to 37˚C for 20 minutes in an incubator. A NanoVueTM spectrophotometer (Harvard Bioscience Inc., Holliston, Massachusetts, USA) was used for Hb evaluation by adding 15 μl of the sample into the reagent and was read at 405 nm in a single measurement mode.

A total of 80 μl of the color reagent mixture provided by the manufacturer in the OSMMR kit was then added to the sample. Following mixing, the microplate was once more read in kinetic mode at 550 nm and readings were taken at 0 and 15 minutes. The total change in

Table 1. Primers design sets for molecular analysis of G6PD mutation.

| Target exon | Primer design (5' to 3') | Annealing Temperature (°C) | PCR product size (bp) |
|---|---|---|---|
| 9, 10 | F: CCT CAA CCC CGG AGA AGT CA<br>R: TGA AGA ACA TGC CCG GCT TC | 56.0 | 869 |
| 11, 12, 13, and 3' untranslated region (UTR) | F: ACG TGA AGC TCC CTG ACG C<br>R: CCA TGG AGT GCA GAG TTG GT | 64.5 | 965 |
| 1 | F: TAA AAA CAC AAG CCC CGC CC<br>R: CTC AAG CAC AAC AAA CAG CGT | 4.4 | 900 |
| 2 | F: GAA TAC ACC AAT GCT TTG AGT<br>R: GCT CAA CTT AGC AGA GCC TGT | 56.0 | 489 |
| 3, 4 | F: TCG GGG CTC TTC TGT CTG TA<br>R: GCT GGT AAT GGG GGT CTC AA | 61.4 | 558 |
| 5 | F: TGT CTC CCA GGC CAC CCC AGA G<br>R: GAC ACG CTC ATA GAG TGG TG | 64.5 | 305 |
| 6 | F: GAG GAG GTT CTG GCC TCT ACT<br>R: AGA TCC TGT TGG CAA ATC TGC AG | 55.6 | 464 |
| 7, 8 | F: GAC AAG GGT GAC CCC TCA CA<br>R: CTG TGC TCA GAG GTG GTG ACT T | 64.4 | 793 |

optical density was calculated from the readings and the final results were expressed in IU/gHb. The adjusted male median (AMM) was determined, and the value was defined as 100% G6PD activity. The 30% and 70% cut-off values were established and used for the diagnosis of G6PD deficiency among the population. These cut-off values were based on the recent World Health Organization (WHO) recommendation for phenotypic classification of G6PD deficiency, as well as corresponded to the safe cut-off points for receiving antimalarial treatment including Primaquine and Tafenoquine. This approach was also adopted by the majority of comparable studies in previous years [21,22].

For all G6PD-deficient samples with activities below 70% of AMM and a subset of G6PD-normal as identified by the G6PD OSMMR2000-D kit assay (n = 69), molecular analysis was performed using the DNA sequencing method to characterize their mutations. The genomic DNA of these samples was extracted from peripheral blood leucocytes using the QIAamp DNA Blood Mini Kit (Qiagen Diagnostics GmbH, Hilden, Germany). Then, the extracted DNA was amplified through the polymerase chain reaction (PCR) technique using specific primer sets (Table 1) with specific PCR conditions (Table 2).

The PCR products were then purified using the QIAquick PCR Purification Kit (Qiagen Diagnostics GmbH, Hilden, Germany) followed by a quality and purity analysis using a similar spectrophotometer. An automated Sanger DNA sequencing was performed on the purified PCR products by services from Apical Scientific Pvt. Ltd (previously known as First BASE Laboratories Pvt. Ltd, Selangor, Malaysia).

Table 2. Conditions for polymerase chain reaction (PCR) technique.

| Temperature (°C) | Steps | Duration (s) | Cycle |
|---|---|---|---|
| 95.0 | Initial denaturation | 300 | 1 |
| 98.0 | Denaturation | 60 | |
| Refer to Table 1 | Annealing | 60 | 30 |
| 72.0 | Extension | 60 | |
| 72.0 | Final extension | 600 | 1 |
| 10.0 | Storage | Infinite | - |

## Data analysis

All data were collated and analyzed using Microsoft® 365 Excel Spreadsheet Software (Microsoft Corporation, WA, USA). The adjusted male median (AMM) was calculated from all male participants and defined as 100% G6PD activity. The method for AMM calculation was described by Ley et al. (2017) [23]. From the AMM, 30% and 70% cut-off values were established to classify the G6PD status of the participants. The overall prevalence of G6PD deficiency was calculated by identifying and excluding all related family members. The prevalence among male and female participants was determined and compared accordingly. The results were statistically analyzed using IBM SPSS Statistics 26.0 for Windows. The results of the molecular analysis were analyzed using Applied Biosystem Sequence Scanner Software v2.0 (Thermo Fisher Scientific, Massachusetts, USA).

## Results

### Study population and distribution of G6PD activity

All 662 venous blood samples of the Senoi Orang Asli were analyzed for G6PD activity (293 males and 369 females). The participants were healthy, afebrile, and free of malaria. There were 102 children aged from 2 to 12 years old, while the remaining 473 samples were adults aged from 13 to 79 years old. The major subethnicities of participants were Temiar (201; 30.3%), followed by Semai (186; 28.1%), and Jah Hut (167; 25.2%), the remaining were Semoq Meri (53; 8.0%), Mah Meri (47; 7.0%) and Che Wong (8; 1.2%).

The AMM of the studied population was 11.8 IU/gHb. Individuals with enzyme levels less than 30% of AMM were classified as deficient, while those with enzyme levels between 30% to 70% were classified as intermediate deficiency. G6PD-normal participants were those with enzyme activity of more than 70% of AMM. Table 3 shows the corresponding values for each phenotypic classification for this study.

The prevalence of G6PD deficiency was 15.2% (95% Confidence Interval: 11–19%; 56/369), with males (30/172; 17.4%) outnumbering females (26/197; 13.2%). A total of 36 participants (9.6%; 26 male and 10 female) were G6PD-deficient (<30% of AMM) and 20 participants (5.4%; 4 male and 16 female) were G6PD-intermediate (30–70% of AMM). A Chi-square test was performed to compare the proportions of deficient and intermediate individuals between females and males. Results revealed a statistically significant higher proportion of deficient individuals in males as compared to females (p = 0.000459) and a statistically significant higher proportion of intermediate individuals in females as compared to males (p = 0.007). These results with their enzyme levels were summarized in Table 4.

### The spectrum of G6PD mutation

Molecular analysis was performed on 87 samples including all G6PD-deficient and G6PD-intermediate blood samples (n = 56) and a subset of G6PD-normal samples (n = 31). Our analysis revealed 27 hemizygote males, 18 heterozygote females, 7 homozygote females, and 2 compound heterozygote females. A total of 18 genotyped samples showed no mutation (Fig 1).

**Table 3. Classification of G6PD activity in the Senoi Malaysian Orang Asli.**

| Measurement Parameter | G6PD activity (IU/gHb) |
|---|---|
| Adjusted Male Median (AMM) (100% activity) | 11.8 |
| G6PD-deficient (<30% of AMM) | < 3.6 |
| G6PD-intermediate (30–70% of AMM) | 3.6–8.3 |
| G6PD-normal (>70% of AMM) | >8.3 |

**Table 4. Prevalence and distribution of G6PD deficiency among the Senoi Malaysian Orang Asli population.** (n = 369).

| Gender | Phenotype | Prevalence (N, %) | G6PD activity (IU/gHb) | |
|---|---|---|---|---|
| | | | Range | Mean (±SD) |
| Male | Deficient | 26 (7.0%) | 0.7–3.3 | 2.0 (±0.9) |
| | Intermediate | 4 (1.1%) | 4.0–8.2 | 6.1 (±2.3) |
| | Normal | 142 (38.5%) | 8.4–22.0 | 12.2 (±2.5) |
| Female | Deficient | 10 (2.7%) | 0.7–3.5 | 1.5 (±0.9) |
| | Intermediate | 16 (4.3%) | 3.7–9.3 | 6.0 (±1.7) |
| | Normal | 171 (46.3%) | 9.4–23.3 | 12.9 (±2.9) |
| Overall | Deficient | 36 (9.8%) | | |
| | Intermediate | 20 (5.4%) | | |
| | Normal | 313 (84.8%) | | |
| Total | | 56/369 (15.2%) (95% CI: 11–19%) | | |

Seven variants of G6PD mutations were found in 79.3% of the samples (69; 35 males and 34 females). The *IVS11 T93C* mutations were the highest among them (47.1%; n = 41), followed by *rs1050757 (3'UTR +357A>G)* (39.1%; n = 34), *G6PD Viangchan (c.871G>A)* (25.3%; n = 22), *G6PD Union (c.1360C>T)* (21.8%; n = 19), *c.1311C>T* (20.7%; n = 18), *G6PD Kaiping (c.1388G>A)* (8.0%; n = 7), and *G6PD Coimbra (c.592C>T)* (2.3%; n = 2). The genotype of each G6PD mutation and the distribution of silent mutation and polymorphism among the Senoi Malaysian Orang Asli population were summarized in Tables 5 and 6.

## Discussion

G6PD deficiency is a public health concern in Malaysia. One of the most serious clinical consequences of this hereditary disease, particularly in newborns, is neonatal hyperbilirubinemia, which requires prompt diagnosis and treatment to prevent kernicterus, an irreversible bilirubin-induced brain damage [22]. Moreover, the use of anti-malarial treatments such as Primaquine in G6PD deficient patients would increase the risk of acute hemolysis [23]. Therefore, based on the WHO guideline to support the treatment of malaria, it was recommended to screen the target population for G6PD deficiency and to ensure the treatment of jaundice for any population with a prevalence of G6PD deficiency greater than 3% among males [24].

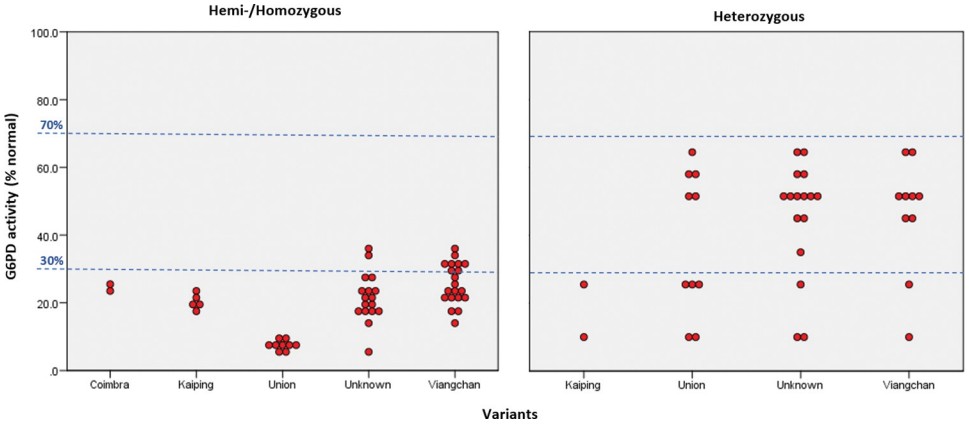

**Fig 1. G6PD activity distributions (% Normal) for Coimbra, Kaiping, Union, Viangchan, and unknown variants among the Senoi Malaysian Orang Asli population.**

**Table 5. G6PD genotypes of the Senoi Malaysian Orang Asli population.**

| G6PD variant | Total subjects (n) | Genotypes (n) | G6PD activity (IU/gHb) | | |
|---|---|---|---|---|---|
| | | | Range | Mean (±SD) | 95% CI |
| *G6PD Coimbra (c.592 C>T)* | 2 | Hemizygote (2) | 2.8–2.9 | 2.8 ±0.02 | 2.7–3.0 |
| *G6PD Viangchan (c.871 G>A)* | 21 + 1[a] | Hemizygote (13) | 1.6–4.2 | 2.8 ±0.71 | 2.4–3.2 |
| | | Heterozygote (9) | 1.2–7.8 | 5.6 ±0.91 | 4.9–6.3 |
| | | Homozygote (5) | 2.6–3.8 | 3.5 ±0.52 | 2.8–4.1 |
| | | Compound heterozygote (1[a]) | 2.9 | | |
| *G6PD Kaiping (c.1388 G>A)* | 6 + 1[b] | Hemizygote (5) | 2.1–2.8 | 2.4 ±0.28 | 2.1–2.6 |
| | | Heterozygote (1) | 2.9 | | |
| | | Compound heterozygote (1[b]) | 1.4 | | |
| *G6PD Union (c.1360 C>T)* | 17 + 1[a] +1[b] | Hemizygote (7) | 0.7–1.1 | 0.9 ±0.11 | 0.8–1.0 |
| | | Heterozygote (8) | 3.1–7.7 | 6.3 ±1.41 | 5.0–7.5 |
| | | Homozygote (2) | 0.7–1.0 | 0.8 ±0.17 | 0–2.7 |
| | | Compound heterozygote (1[a] + 1[b]) | | | |
| Unknown variants | 33 | Male (17) | 0.6–4.0 | | |
| | | Female (16) | 0.9–7.5 | | |

1[a] A compound heterozygote case of G6PD Viangchan + Union.

1[b] A compound heterozygote case of G6PD Kaiping + Union.

The detection of G6PD deficiency was significantly higher in this study compared to previous reports. The higher detection of G6PD deficiency in our study were owing to the use of a quantitative G6PD activity assay, the OSMMR-2000D kit assay, rather than the widely used semiquantitative FST. In numerous earlier research, the OSMMR-2000D kit assay served as the gold standard quantitative G-6-PD enzymatic test. [18,19,25–28]. In contrast, the FST has been shown to perform poorly in detecting partial G6PD deficiency among female heterozygotes, with up to 14.8% of subjects misdiagnosed as normal [18].

In regard to the genetic composition, G6PD deficiency exhibits high ethnogeographic variability across the global population [29]. Previous studies proved that each ethnic group in Malaysia has a unique spectrum of G6PD mutations. As an example, in Malay, the most common G6PD variant discovered was *G6PD Viangchan (c.871 G>A)*, accounting for 37.2% of the deficient cases, followed by *G6PD Mediterranean (c.563C>T;* 26.7%), and *G6PD Mahidol (c.487G>A;* 15.1%) [3]. Other mutation variants with lower prevalence identified in Malay were *G6PD Canton (c.1376G>T)*, *G6PD Chatham (c.1003G>A)*, and *G6PD Andalus (c.1361G>A)*. This broad spectrum of mutation illustrated the Malay people's rich history of interaction with neighboring Southeast Asia ethnic groups, outside traders, and Chinese and Indian immigrants [3].

**Table 6. Distribution of silent mutation and polymorphism among the studied population.**

| Gender | Subethnicity | *c.1311 C>T* | *IVS1193 T>C* | *3'UTR +357 A>G* |
|---|---|---|---|---|
| Male | Jah Hut | 1 | 6 | 6 |
| | Semai | 6 | 10 | 10 |
| | Temiar | 0 | 3 | 0 |
| Female | Jah Hut | 2 | 6 | 5 |
| | Semai | 9 | 12 | 12 |
| | Temiar | 0 | 4 | 1 |
| Total | | 18 | 41 | 34 |

Meanwhile, for Chinese ethnicity in Malaysia, *G6PD Canton* (*c.1376G > T*) and *G6PD Kaiping* (*c.1388 G>A*) were the most common variants, representing 83.6% of cases. Several other Chinese-specific variants with lower frequencies identified were *G6PD Gaohe* (*c.95A>G*), *G6PD Nankang* (*c.517T>C*), *G6PD Chinese-5* (*c.1024C>T*), and *G6PD Quingyuan* (*c.392G>T*) [4,30]. The finding of multiple variants among the Chinese ethnicity reflected the historical origins of the early Chinese ancestors who immigrated to Malaysia. However, there was a lower prevalence of G6PD deficiency among the Indian ethnicity, where some individuals were found to carry different variants including *G6PD Namoru* (*c.208T>C*) and *G6PD Mediterranean* (*c.563C>T*) [14].

In contrast, information and studies on the G6PD status and mutations of the indigenous Malaysian Orang Asli are scarcer. In a prior investigation, Wang et al. (2008) discovered a few cases of *G6PD Coimbra* (*c.592 C>T*), *G6PD Viangchan* (*c.871 G>A*), *c.1311C>T*, and *IVS11 93 T>C* in the Orang Asli [15]. Our previous study on five Negrito subethnicities of the Orang Asli in 2011 found that Lanoh had the highest prevalence (28%), followed by Kintak (18%), Bateq (15%), Jahai (3%), and none in Kensiu [16]. *G6PD Viangchan*, *G6PD Coimbra*, and *rs1050757* make up only a small spectrum of the Negrito, in comparison to the Malay and Chinese. Despite the earlier presence of Orang Asli in Peninsular Malaysia, we believe this smaller and less diversified mutation was caused by the profound consequences of inbreeding homophily [20].

In this present study, we confirmed the high prevalence of G6PD deficiency among the Senoi Malaysian Orang Asli, with a significant degree of molecular heterogeneity. In comparison with the prevalence of the Negrito (9%), Senoi had a higher prevalence of 15.2% [17]. Therefore, according to this finding, Senoi has the highest rate of G6PD deficiency in Malaysia. According to our prevalence study, most of the G6PD-deficient Senoi Orang Asli population had G6PD deficiency with enzyme activity less than 30% of AMM (9.8%, 36/369), while only 5.4% (20/369) of the subjects had intermediate deficiency with enzyme activity between 30% to 80% of AMM.

Seven mutations were found in 69 out of 87 genotyped samples including G6PD *Coimbra (c.592C>T)*, *G6PD Kaiping (c.1388G>A)*, *G6PD Union (c.1360C>T)*, *G6PD Viangchan (c.871G>A)*, *c.1311C>T*, *IVS11 93 T>C*, and the *3' UTR mutation +357A>G (rs1050757)* polymorphism. The most common variant among the Senoi Orang Asli was G6PD *Viangchan (c.871 G>A)* with 22 cases (31.8%) including a case of compound female heterozygote of *G6PD Viangchan* and *G6PD Union* variants. The *G6PD Kaiping* variant was found only among the Jah Hut subethnicity. The exclusivity of this variant supported the idea that this subethnic group had Chinese ancestry [31]. The Senoi population also had 19 cases of the *G6PD Union* variant. This variant was predominantly found in the Philippines and Papua New Guinea populations, as well as in Thailand, Vietnam, and the Solomon Islands [30–34]. This molecular discovery suggested a direct link between this subethnic group and the people of Indochina.

Finally, silent mutation and polymorphism were identified in three different subethnicities of the Senoi Orang Asli, the Jah Hut, Semai, and Temiar. For the *3' UTR mutation +357A>G (rs1050757)* polymorphism, it was discovered in 39.1% (34) of the G6PD deficient individuals, including 18.4% (16) in males and 20.9% (18) in females. Amini et al (2013) had previously reported this mutation in 85.4% of the G6PD-deficient Negrito Orang Asli group, with a strong association with haplotype *1311T/IVS11 93 C* [20]. The finding revealed the possibility of intermarriage between the Senoi and the Negrito subethnic groups. However, more expression studies are warranted to ascertain the role of the *rs1050757* mutation in the epidemiology of G6PD deficiency in Senoi and Negrito populations. In order to complete the genetic picture of G6PD deficiency in the Malaysian Orang Asli, we recommend molecular screening among

the Proto-Malay group of Orang Asli in future studies. By comparing the results and annotating the variants from all groups and ethnicities, we hope to improve G6PD health care and malaria eradication for the Orang Asli population.

## Conclusions

We confirmed that the Senoi Malaysian Orang Asli subethnic group has the highest prevalence of G6PD deficiency in Malaysia, with a significant difference in its molecular variants. Our findings highlight the importance of determining G6PD status in all Malaysian Orang Asli populations, for a greater emphasis on the proper and safe use of Primaquine for malaria elimination.

## Supporting information

**S1 File. G6PD activity of Senoi Malaysian Orang Asli.** The study's minimal underlying data set.
(PDF)

## Acknowledgments

We would like to thank the National University of Malaysia, Universiti Kebangsaan Malaysia (UKM) Medical Centre, and the Department of Orang Asli Development (JKOA) for their approval to conduct this research. Many thanks to all research participants, co-investigators, and laboratory personnel for their hard work on this project.

## Author Contributions

**Conceptualization:** Danny Xuan-Rong Koh, Ainoon Othman, Endom Ismail.

**Data curation:** Danny Xuan-Rong Koh, Nur Awatif Akmal Muhamad Hata, Ainoon Othman, Endom Ismail.

**Formal analysis:** Danny Xuan-Rong Koh, Raja Zahratul Azma Raja Sabudin, Sanggari Muniandy, Endom Ismail.

**Funding acquisition:** Raja Zahratul Azma Raja Sabudin.

**Investigation:** Danny Xuan-Rong Koh, Sanggari Muniandy, Nur Awatif Akmal Muhamad Hata, Siti Noor Baya Mohd Noor, Norhazilah Zakaria.

**Methodology:** Danny Xuan-Rong Koh, Siti Noor Baya Mohd Noor, Norhazilah Zakaria, Ainoon Othman, Endom Ismail.

**Project administration:** Raja Zahratul Azma Raja Sabudin, Ainoon Othman.

**Resources:** Raja Zahratul Azma Raja Sabudin.

**Supervision:** Raja Zahratul Azma Raja Sabudin, Ainoon Othman, Endom Ismail.

**Validation:** Danny Xuan-Rong Koh, Mohamed Afiq Hidayat Zailani, Raja Zahratul Azma Raja Sabudin, Ainoon Othman, Endom Ismail.

**Writing – original draft:** Danny Xuan-Rong Koh.

**Writing – review & editing:** Mohamed Afiq Hidayat Zailani, Raja Zahratul Azma Raja Sabudin, Ainoon Othman, Endom Ismail.

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
