## [Decision Letter · Decision Letter 0]

13 Mar 2023

PONE-D-23-01672Prevalence and molecular heterogeneity of glucose-6-phosphate dehydrogenase (G6PD) deficiency in the Senoi Malaysian Orang Asli populationPLOS ONE

Dear Dr. Raja Sabudin,

Thank you for submitting your manuscript to PLOS ONE. After careful consideration, we feel that it has merit but does not fully meet PLOS ONE’s publication criteria as it currently stands. Therefore, we invite you to submit a revised version of the manuscript that addresses the points raised during the review process.

We look forward to receiving your revised manuscript.

Kind regards,

Germana Bancone, Ph.D

Academic Editor

PLOS ONE

Journal Requirements:

    "This study was funded by the Ministry of Higher Education (MOHE), Malaysia through the Exploratory Research Grant Scheme (Reference Code: ERGS/1/2012/STG03/UKM/02/1). We would like to thank the Universiti Kebangsaan Malaysia (UKM) Medical Centre and the Department of Orang Asli Development (JKOA) for their approval to conduct this research. Many thanks to all research participants, co-investigators, and laboratory personnel for their hard work on this project"

  "RZA received the award that funded this study. This study was funded by the Ministry of Higher Education (MOHE), Malaysia through the Exploratory Research Grant Scheme (Grant number: ERGS/1/2012/STG03/UKM/02/1). The website of funder is https://www.mohe.gov.my/. The funders had no role in study design, data collection and analysis, decision to publish, or preparation of the manuscript."

Additional Editor Comments:

Please provide the full data supporting this manuscript

Reviewers' comments:

Reviewer's Responses to Questions

**Comments to the Author**

1. Is the manuscript technically sound, and do the data support the conclusions?

Reviewer #1: Partly

Reviewer #2: Yes

2. Has the statistical analysis been performed appropriately and rigorously? 

Reviewer #1: No

Reviewer #2: Yes

3. Have the authors made all data underlying the findings in their manuscript fully available?

Reviewer #1: No

Reviewer #2: No

4. Is the manuscript presented in an intelligible fashion and written in standard English?

Reviewer #1: No

Reviewer #2: Yes

5. Review Comments to the Author

Reviewer #1: Kindly see the attached form that has a better formatting

Review of the manuscript: “Prevalence and molecular heterogeneity of glucose-6-phosphate dehydrogenase (G6PD) deficiency in the Senoi Malaysian Orang Asli population” (PONE-D-23-01672)

Summary: This article reports the results of a survey among one of Malaysia’s ethnicity (Senoi Malaysian Orang Asli) on the prevalence and dominant genotypes of G6PD deficiency. A total of 662 participants were screened by phenotypic test and all those identified as deficient (n=87, 13%) were genotyped. Among all phenotypic deficient participants, seven G6PD variants were identified. The authors conclude that the prevalence of G6PD deficiency was very high among the study population.

General: The addressed question is relevant for the local context. My key concerns are:

• The authors report overall prevalence of G6PD deficiency within the target population but do not provide any details on how participants were recruited. If participants were not recruited at random and multiple members of the same family were enrolled, the reported prevalence will be biased. The authors need to report prevalence together with its variation (95% confidence interval or interquartile range)

• The authors do not explain their statistical analysis in the methods section, some information is provided in the results section that needs to be moved accordingly. From the results section I understand that the authors calculated the mean G6PD activity after excluding everyone with activities below the manufacturers recommended range of normal activities (<7.8U/gHb). This mean was then defined as 100% activity and anyone with <10% activity was categorised as severe and anyone with >10% activity and <60% was defined as moderate deficient. I recommend that the authors calculate the adjusted male median (AMM) [1] instead and define anyone with activities below 30% of the AMM as deficient and anyone with activities >30% and <70% of the AMM as having intermediate activities. This is the approach chosen by the majority of comparable articles (for example [2]). I am also unsure why the authors stratify their analysis by age group? I am not aware that G6PD activity would differ between 2–12-year-old and older individuals? The here observed difference is likely due to chance.

• Genotyping only phenotypically deficient individuals introduces a bias. Is there any chance the authors can also genotype a random subset of phenotypically G6PD normal individuals?

Minor comments:

• Can the authors add definitions of G6PD deficiency in the abstract?

• Can the authors add total numbers of males and females enrolled in the abstract to put the proportion of deficinet males and females into perspective?

• In the abstract pervalence of G6PD deficinecy is mentioned twice, in lines 32 and 34, kindly streamline

• In the abstract, can the authors start of by stating that no mutation was found among 18 genotyped and deficnet indidvausl (if I counted correct).

• In the abstract can the authors include how many individuals were found to be hemi/homo or heterozygous?

• Line 38 and following, can the authors add absolute numbers to the proportions?

• Line 48: this statement requires a reference.

• Line 52: replace “disease” by “condition”?

• Line 53: G6PD deficient indivduals can be found in almost all populations, not only the mentioned continents.

• Line 74 states that Primaquine is the most widely used antimalarial and this statement will need a reference.

• Line 77 states: “Despite the high susceptibility and endemicity of malaria among the Orang Asli in Malaysia” – can the authors clarify what susceptoibility refer to?

• Table 3: can the authors add that this table only covers G6PD normals? Why not include deficients as a comparisson as well?

• Line 173: how was the level of significance calculated? Please add this to the methods.

• Table 5: rather than categorising all variants as “moderate” or “severe”, present range of observed activities / variant

• Table 5: In the “total: column there is a mention of “21+2” etc. and it is unclear what this means. Is there a footnote missing?

• Throughout the text the authors refer to a variation as ± (for example 14.33±3.48 in line 36) and this is ambiguous. Can the authors replace this by mean and 95%CI for normal distributed data and median and IQR for not normally distributed data?

• Were all partricipanst afebrile and free of malaria? If yes, kindly add.

• Throughout the entire article, the authors should include manufacturer and country of the manufacturer whenever mentioning a product.

Language: the language of the manuscript must be revised by a native English speaker familiar with the topic

Reference:

1. Domingo, G.J., et al., G6PD testing in support of treatment and elimination of malaria: recommendations for evaluation of G6PD tests. Malar J, 2013. 12: p. 391.

2. Ley, B., et al., Wide range of G6PD activities found among ethnic groups of the Chittagong Hill Tracts, Bangladesh. PLOS Neglected Tropical Diseases, 2020. 14(9): p. 8697--1.

Reviewer #2: This is a well written paper with clear and concise findings. I only have minor edits to suggest before moving this forward to publication.

Ln 24 italicize plasmodium

Ln 31 subethnic group

Ln 76 P. falciparum

Were the quantitative assays run in duplicate? I am unfamiliar with this kit, it might be valuable to include any literature comparing this kit with Point scientific or trinity biotech G6PD spectrophotometric tests.

General comment – be consist with use of sub-ethnic vs subethnic - same goes for subethnicity

How did normal G6PD thresholds match up with the study by Pfeffer et al. “Quantification of glucose-6-phosphate dehydrogenase activity by spectrophotometry: A systematic review and meta-analysis” I’d be interested in seeing if they aligned, given the unique population of this study.

6. PLOS authors have the option to publish the peer review history of their article (what does this mean?). If published, this will include your full peer review and any attached files.

Reviewer #1: No

Reviewer #2: No

---

## [Author Response · Author response to Decision Letter 0]

4 Sep 2023

Response to Editor's comments:

1. Thank you for the comments. We have checked though the PLOS ONE's style requirements and made necessary amendments, including the file names.

2. We apologize for the tehcnical issue. We have now removed the funding-related text from the manuscript (the Acknowledgement section). Thank you, we agreed with the current Funding Statement and no further updates are required from our side.

3. We have uploaded our study’s minimal underlying data set as Supporting Information file 1 (S1_file.pdf). We have fully anonymized the data and removed any potentially identifying patient information. Thank you.

Response to Reviewer 1:

1. Thank you for your valuable comments. The authors have exluded the family members in calculation of the overall prevalence to prevent bias. We have recalculated and provided a revised prevalence, as well as its variation (95% confidence interval). These results were written in abstract as well as in the body of the manuscript.

In the abstract:

“…The overall prevalence of G6PD deficiency was 15.2% (95% Confidence Interval: 11-19%; 56 of 369) , with males (30 of 172; 17.4%) outnumbering females (26 of 197; 13.2%). The adjusted male median (AMM), defined as 100% G6PD activity, was 11.8 IU/gHb. A total of 36 participants (9.6%; 26 male and 10 female) were severely deficient (<30% of AMM) and 20 participants (5.4%; 4 male and 16 female) were G6PD-intermediate (30 - 80% of AMM).”

2. Thank you for this comment. We have added the explaination on statistical analysis in the methods section (Data analysis) as suggested.

We appreciate the Reviewer’s recommendation. We have amended our calculation using the adjusted male median (AMM). We have also used the cut-off threshold of 30% and 70% as per advised.

We agreed with the comments given regarding the age. We have amended our manuscript and did not stratify the analysis by age group.

3. We have genotyped a random subset of phenotypically G6PD normal individuals. We have now included this point in our manuscript.

“Molecular analysis was performed on 87 samples including all G6PD-deficient blood samples (n=56) and subset of G6PD-normal sample (n=31).”

MINOR

Thank you for the excellent comments. We have amended our manuscript accordingly:

1. We have added definitions of G6PD deficiency in the abstract.

“Glucose-6-phosphate dehydrogenase (G6PD) deficiency is an X-linked genetic disorder characterized by reduced G6PD enzyme level in the blood.”

2. We have added the total numbers of males and females enrolled in the abstract.

“…A total of 662 blood samples (369 males and 293 females).”

3. We have amended the lines mentioned and streamlined the details.

4. Excellent suggestion. We have included this point as suggested. Yes, the number given (n=18) was correct.

“A total of 87 samples were genotyped, of which 18 showed no mutation.”

5. We have included the number of participants with the hemi/homo or heterozygous in the abstract as suggested.

“Our analysis revealed 27 hemizygote males, 18 heterozygote females, 7 homozygote females, and 2 compound heterozygote females.”

6. In lines 38 (abstract), the authors have added absolute numbers to the proportions as requested.

“…Seven mutations were found among 69 genotyped samples; IVS11 T93C (47.1%; n=41), rs1050757 (3’UTR +357A>G)(39.1%; n=34), G6PD Viangchan (c.871G>A)(25.3%; n=22), G6PD Union (c.1360C>T)(21.8%; n=19), c.1311C>T(20.7%; n=18), G6PD Kaiping (c.1388G>A)(8.0%; n=7), and G6PD Coimbra (c.592C>T)(2.3%; n=2).

7. The authors have added a suitable reference for this statement.

Luzzatto L, Ally M, Notaro R. Glucose-6-phosphate dehydrogenase deficiency. Blood. 2020;136(11): 1225–1240. doi: 10.1182/blood.2019000944

8. We have replaced the word “disease” with “condition”.

9. We agreed with this comment, and we have amended the sentence accordingly to address this point.

“This X-linked hereditary condition affects the entire world, but it is more prevalent particularly in parts of the African continent, the Middle East, and Southeast Asia, where it often overlaps with the geographical distribution of malaria infection.”

10. We have added a reference for this statement.

Camarda G, Jirawatcharadech P, Priestley RS, Saif A, March S, Wong MH, Leung S, et al. Antimalarial activity of primaquine operates via a two-step biochemical relay. Nature communications. 2019 Jul 19;10(1):3226.

11. We have added a sentence to further clarify the point of high susceptibility.

“…The Orang Asli population is highly susceptible to mosquito bites and malaria infection due to their isolated settlements in tropical forests and traditions of hunting and foraging for food in the jungle….”

12. Thank you for this excellent comment. We have amended the table accordingly (Refer Tables 3 and 4).

13. We have added the method for calculation of significance in the “Data Analysis” section.

14. We have amended the table and presented the range of observed activities as suggested (Refer Table 5).

15. We have added the footnote missing from the table.

“Footnote: 1a refers to a compound heterozygote case of G6PD Viangchan + Union, and 1b refers to a compound heterozygote case of G6PD Kaiping + Union”

16. Thank you for the comment. We have included the 95%CI for our AMM value and G6PD levels for all genotypes in Table 5. However, for other values throughout the text, we would like to respectfully propose using the mean and standard deviation (SD) (example: 14.33±3.48 ) as measures of variation, rather than the 95% confidence interval (CI), to represent our data. We believe that the mean and SD provide a clear and concise way to describe the central tendency and spread of our data, without the additional complexity introduced by the CI. The use of these statistics aligns with common reporting practices in our field and maintains the readability of our manuscript.

17. Yes, all participants were afebrile and free of malaria. We have added this information in the “Result” section.

“The participants were healthy, afebrile, and free of malaria.”

18. We have checked the entire manuscript and included the details of the manufacturer and country of the manufacturer for all products mentioned. All details were provided in the first mention, and these details were not included in subsequent mention throughout the manuscript (aligned with the journal’s guideline).

For language improvement, we have conducted a thorough grammar check and revision of the manuscript to ensure its clarity and readability. We have included the relevant references given in our reference list.

Response to Reviewer 2:

1. Thank you. We have amended all mentioned points accordingly, including Lines 24,31,76, as suggested.

2. The quantitative assays were not run in duplicate for our study. While trying our best, we have included a statement in addresing this point with inclusion of several relevant references.

“In numerous earlier research, the OSMMR-2000D kit assay served as the gold standard quantitative G-6-PD enzymatic test. [18,19,25 – 28].”

3. We have checked through the manuscript and be consistent in using subethnic and subethnicity (13 replacements).

4.Thank you for your high interest. We appreciate your comment.

In our study, the normal threshold for 100% G6PD activity corresponded to a slightly higher value of 11.8 IU/gHb. The mentioned paper by Pfeffer et al. (2020) stated that a universal threshold of 100% G6PD activity was defined as 9.4 IU/gHb. However, the authors also highlighted that caution is stricly advised in comparing findings based on absolute G6PD activity measurements across studies due to variability in laboratory methods, with possible contribution of unmeasured population factors. We have briefly included this interesting point and the mentioned reference in our manuscript as well.

“The normal threshold for 100% G6PD activity, the AMM, of our study corresponded to a slightly higher value of 11.8 IU/gHb compared to a universal threshold of 9.4 IU/gHb, as described by a meta analysis study of Pfeffer et al. (2020) [25]. However, as described by the article, researchers should strongly cautioned against drawing conclusions solely on absolute G6PD activity from different studies, due to interlaboratory variability and possible unmeasured population factors.”

---

## [Decision Letter · Decision Letter 1]

18 Sep 2023

PONE-D-23-01672R1Prevalence and molecular heterogeneity of glucose-6-phosphate dehydrogenase (G6PD) deficiency in the Senoi Malaysian Orang Asli populationPLOS ONE

Dear Dr. Raja Sabudin,

Thank you for submitting your manuscript to PLOS ONE. After careful consideration, we feel that it has merit but does not fully meet PLOS ONE’s publication criteria as it currently stands. Therefore, we invite you to submit a revised version of the manuscript that addresses the points raised during the review process.

We look forward to receiving your revised manuscript.

Kind regards,

Germana Bancone, Ph.D

Academic Editor

PLOS ONE

Additional Editor Comments:

Please address the reviewer's comments

Reviewers' comments:

Reviewer's Responses to Questions

**Comments to the Author**

1. If the authors have adequately addressed your comments raised in a previous round of review and you feel that this manuscript is now acceptable for publication, you may indicate that here to bypass the “Comments to the Author” section, enter your conflict of interest statement in the “Confidential to Editor” section, and submit your "Accept" recommendation.

Reviewer #1: (No Response)

Reviewer #2: All comments have been addressed

2. Is the manuscript technically sound, and do the data support the conclusions?

Reviewer #1: Yes

Reviewer #2: Yes

3. Has the statistical analysis been performed appropriately and rigorously? 

Reviewer #1: No

Reviewer #2: Yes

4. Have the authors made all data underlying the findings in their manuscript fully available?

Reviewer #1: Yes

Reviewer #2: Yes

5. Is the manuscript presented in an intelligible fashion and written in standard English?

Reviewer #1: Yes

Reviewer #2: Yes

6. Review Comments to the Author

Reviewer #1: Please see attachment for better formatting

Re-Review of the manuscript: “Prevalence and molecular heterogeneity of glucose-6-phosphate dehydrogenase (G6PD) deficiency in the Senoi Malaysian Orang Asli population” (PONE-D-23-01672_R1)

Summary: Most of my comments have been addressed (many thanks), I have some comments around the involved statistics, kindly see below, line numbers refer to the manuscript with highlighted changes.

Statistics:

• Line 164 to 168. Why do the authors calculate the mean G6PD activity and the AMM? This is a bit confusing. It would be easier to just calculate and report the AMM.

• Line 168: the authors describe how they assessed differences in G6PD activities using a paired T-test and this is the wrong test. A paired test requires paired measurements (for example by testing the same person with two different assays) but I don’t think this is given here. Also, G6PD activity is probably not normally distributed, and this is one of the requirements for a T-test. The authors should consider using the Mann-Whitney U test instead.

• Line 195: The authors report the mean G6PD activity for children and adults stratified. I cannot understand how this distinction could be useful (please also see my previous comments). Unless the current literature is wrong, G6PD activity does not change much once an individual has reached one year of age. Any observed differences between children and adults are therefore likely due to chance. Consider removing this element?

• Line 196 – 197: Rather than comparing mean activities between age groups and gender, it would be better to compare proportions of deficient and intermediate individuals between females and males. The authors could consider a Chi-square test for this.

• Lines 208 – 211: Can the authors show how genotype and phenotype are associated? Perhaps in a figure where G6PD activities for each genotype (and for genotypically unknown/normal participants) is displayed. For an example see [1] figure 2

Minor comments:

• Line 197: Change this sentence so it reads “The prevalence of G6PD deficiency…”

• Line 239: Pfeffer et al do not describe a universal G6PD cut-off, please remove this statement

1. Pfeffer, D.A., et al., Genetic Variants of Glucose-6-Phosphate Dehydrogenase and Their Associated Enzyme Activity: A Systematic Review and Meta-Analysis. Pathogens, 2022. 11(9).

Reviewer #2: (No Response)

7. PLOS authors have the option to publish the peer review history of their article (what does this mean?). If published, this will include your full peer review and any attached files.

Reviewer #1: **Yes: **Benedikt Ley

Reviewer #2: No

---

## [Author Response · Author response to Decision Letter 1]

5 Oct 2023

Statistic:

1. Thank you for your comments. The authors agreed with the point given and have removed the calculation of the mean as suggested. The authors have only reported the AMM.

(Deleted sentence -Line 164-168)

2. Thank you for your insightful comments. The mentioned line (Line 168) reported on mean G6PD activity. This sentence, based on the first remark, has been removed by the authors. As a result, neither the paired T-test nor the Mann-Whitney U test was performed in this revised version.

3. The authors agreed with this comment and have removed this element from the manuscript.

(Deleted sentence -Line 195-197)

4. The authors agreed with this comment and have performed a Chi-square test to compare the proportions of deficient and intermediate individuals between females and males. The following sentences were added:

(Line 191-195)

“A Chi-square test was performed to compare the proportions of deficient and intermediate individuals between females and males. Results revealed no significant difference (p = 0.111) in the proportion of deficient individuals between the two categories. However, there was a statistically significant variation in the proportion of intermediate individuals between males and females in the studied population (p = 0.007).

5. Thank you for this comment. The authors have added a figure to display the association between genotype and phenotype, guided by the given example.

(Fig 1. G6PD activity distributions (% Normal) for Coimbra, Kaiping, Union, Viangchan, and unknown variants among the Senoi Malaysian Orang Asli population.)

Minor comments:

• Line 187: The sentence was amended to “The prevalence of G6PD deficiency was 15.2%”.

• Line 237-242: The statement was removed

---

## [Decision Letter · Decision Letter 2]

13 Nov 2023

Prevalence and molecular heterogeneity of glucose-6-phosphate dehydrogenase (G6PD) deficiency in the Senoi Malaysian Orang Asli population

PONE-D-23-01672R2

Dear Dr. Raja Sabudin,

We’re pleased to inform you that your manuscript has been judged scientifically suitable for publication and will be formally accepted for publication once it meets all outstanding technical requirements.

Kind regards,

Germana Bancone, Ph.D

Academic Editor

PLOS ONE

Additional Editor Comments (optional):

Before publication, I would invite the authors to revise their calculation of Chi squared when comparing G6PD deficient phenotypes between sex. According to my calculation based on numbers provided, both the "severe" phenotype and the intermediate phenotype are significantly different between male and females.

On a correlated note, while defined in the results as activity <30%, the use of term "severe" for G6PD deficiency in this context does not seem appropriate. I would suggest the authors use just "deficient" and "intermediate".

Reviewers' comments:

Reviewer's Responses to Questions

**Comments to the Author**

1. If the authors have adequately addressed your comments raised in a previous round of review and you feel that this manuscript is now acceptable for publication, you may indicate that here to bypass the “Comments to the Author” section, enter your conflict of interest statement in the “Confidential to Editor” section, and submit your "Accept" recommendation.

Reviewer #3: All comments have been addressed

2. Is the manuscript technically sound, and do the data support the conclusions?

Reviewer #3: Yes

3. Has the statistical analysis been performed appropriately and rigorously? 

Reviewer #3: Yes

4. Have the authors made all data underlying the findings in their manuscript fully available?

Reviewer #3: Yes

5. Is the manuscript presented in an intelligible fashion and written in standard English?

Reviewer #3: Yes

6. Review Comments to the Author

Reviewer #3: (No Response)

7. PLOS authors have the option to publish the peer review history of their article (what does this mean?). If published, this will include your full peer review and any attached files.

Reviewer #3: No

---

## [Editor Report · Acceptance letter]

4 Dec 2023

PONE-D-23-01672R2 

Prevalence and molecular heterogeneity of glucose-6-phosphate dehydrogenase (G6PD) deficiency in the Senoi Malaysian Orang Asli population 

Dear Dr. Raja Sabudin:

I'm pleased to inform you that your manuscript has been deemed suitable for publication in PLOS ONE. Congratulations! Your manuscript is now with our production department. 

Kind regards, 

on behalf of

Dr. Germana Bancone 

Academic Editor

PLOS ONE